# Domain Generalization via Invariant Representation under Domain-Class Dependency

## Abstract

Learning domain-invariant representation is a dominant approach for *domain generalization*, where we need to build a classifier that is robust toward domain shifts induced by change of users, acoustic or lighting conditions, etc. However, prior domain-invariance-based methods overlooked the underlying dependency of classes (target variable) on source domains during optimization, which causes the trade-off between classification accuracy and domain-invariance, and often interferes with the domain generalization performance. This study first provides the notion of *domain generalization under domain-class dependency* and elaborates on the importance of considering the dependency by expanding the analysis of Xie et al. (2017). We then propose a method, *invariant feature learning under optimal classifier constrains (IFLOC)*, which explicitly considers the dependency and maintains accuracy while improving domain-invariance. Specifically, the proposed method regularizes the representation so that it has as much domain information as the class labels, unlike prior methods that remove all domain information. Empirical validations show the superior performance of IFLOC to baseline methods, supporting the importance of the domain-class dependency in domain generalization and the efficacy of the proposed method for overcoming the issue.

## 1 Introduction

In supervised learning problems we typically assume that samples are obtained from the same distribution in training and testing; however, such an assumption does not hold in many practical situations, depressing the classification accuracy for the test data (Torralba & Efros (2011)). One typical situation is domain generalization (e.g., Blanchard et al. (2011)): we have labeled data from several source domains and collectively exploit them so that the trained system generalizes to other, unseen but somewhat similar, target domains. Such challenges arise in many applications, e.g., hand-writing recognition (Shankar et al. (2018)), robust speech recognition (Sriram et al. (2018)), and sensor data interpretation (Erfani et al. (2016)).

To address domain generalization, many methods take advantage of invariant feature learning (Muandet et al. (2013); Erfani et al. (2016); Ghifary et al. (2017); Xie et al. (2017)). Such methods assume that learning the representation ($h$) that is invariant to domains ($d$) from input data ($x$) prevents $h$ to overfit to source domains and leads to higher classification accuracy for unseen domains. To obtain such $h$, we used various methods to measure the invariance of $h$ to $d$ and imposed some regularization on the measurement. For example, domain adversarial networks (DAN) (Ganin et al. (2016); Xie et al. (2017)) measure the invariance using a domain classifier (also called a discriminator) parameterized by deep neural networks and impose regularization by deceiving it.

Most prior works, however, overlooked the underlying dependency of classes on source domains, which we refer to as *domain-class dependency*. More specifically, we define domain-class dependency as the situation where domain and class labels are statistically dependent due to some common latent factor ($z$) of $y$ and $d$ (Figure 1-right). Under the domain-class dependency, merely forcing the optimal domain-invariance harms the classification accuracy, as shown in Figure 1-(c). Intuitively speaking, since $y$ contains information about $d$ under domain-class dependency, $h$ must keep at least as much domain information as $y$ to achieve the optimal classification accuracy; however, invariant feature learning attempts to remove all domain information from $h$, which causes the trade-off. It might be similar to the situation where $p(y|x)$ and $p(x)$ change across domains due to the causal

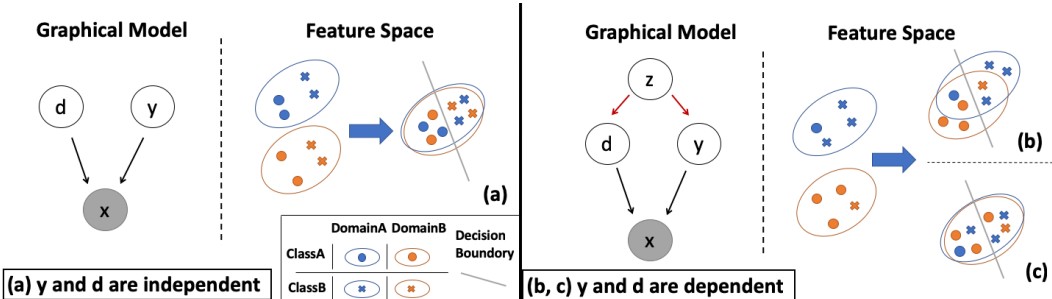

Figure 1: The illustration of the domain-class dependency problem. While Li et al. (2018c) focused on the causal relationship between $x$ and $y$, we focus on the relationship between $y$ and $d$ because it causes the following trade-off problem. (a) When domain and class are independent, domain invariance and classification accuracy can be optimized at the same time. (b,c) In domain-class dependency, there is a trade-off between these two: (b) optimal invariance cannot be achieved when optimal classification accuracy is achieved, and (c) vice versa. We propose a method to lead explicitly to (b) rather than (c), because the primary purpose for domain generalization is classification, not domain-invariance itself.

structure $y \rightarrow x$ (Zhang et al. (2013); Gong et al. (2016) in domain adaptation and Li et al. (2018c) in domain generalization), which we call *conditional probability shift*. However, the shift does not cause the trade-off as long as $y$ and $d$ are independent (Figure 1-left), so it is necessary to focus on the relationship between $y$ and $d$.

Unfortunately, domain-class dependency is common in real-world datasets as shown in Zhang et al. (2013). The dependency can be caused by both the characteristics of data and errors in collecting data. For example, the WISDM Activity Prediction dataset (Kwapisz et al. (2011)), where classes and domains correspond to activities and users, exhibits the dependency because (1) some activities (jogging and climbing stairs) are strenuous (*data characteristics*) and (2) other activities (sitting and standing) and some users were added only after the study began (*data-collection errors*).

In this paper, we address domain generalization under domain-class dependency. We first expand the analysis about DAN by Xie et al. (2017), show that domain-class dependency causes the trade-off problem, and then derive a way to evade the trade-off. Specifically, we investigate the condition where the domain-invariance is maximized under the constraint that it does not interfere with classification accuracy (Figure 1 (b)), because the primary purpose of domain generalization is classification rather than domain-invariance itself. We then propose a novel method *invariant feature learning under optimal classifier constraint (IFLOC)*, modifying DAN's regularization term to make the learned representation have as much domain information as the class labels, i.e., $H(d|h) = H(d|y)$ holds (here $H$ denotes entropy). Like DAN, IFLOC has an encoder, classifier, and domain discriminator, and also takes over the good properties of DAN: it does not depend on pre-defined metrics (e.g., maximum mean discrepancy (Tzeng et al. (2014))), and it can be trained in an end-to-end manner. Empirical validations show the superior performance of IFLOC to baseline methods, supporting the importance of considering domain-class dependency in domain generalization tasks and the efficacy of the proposed approach for overcoming the issue.

The main contributions of this paper can be summarized as follows. Firstly, we elaborate on the trade-off problem under domain-class dependency, both theoretically and experimentally, for the first time in domain generalization context. Secondly, to address the issue we provide theoretical analysis, which shows to what extent latent representations can become invariant to domains without interfering with classification accuracy. Finally, based on the analysis we propose a novel method IFLOC, and validated its efficacy by the experiments on both synthetic and real world datasets.

## 2 RELATED WORKS

Invariant feature learning is a general-purpose method applicable to domain generalization as well as to domain adaptation (e.g., Tzeng et al. (2014); Ganin et al. (2016)), style transfer (e.g., Lample et al. (2017); Chou et al. (2018)), and fairness-aware classification (e.g., Zemel et al. (2013);

Louizos et al. (2016); Madras et al. (2018)). However, it is likely that adjusting it to each specific task can improve performance. For example, in the fairness-aware classification task Madras et al. (2018) proposed to optimize the fairness criterion directly instead of applying invariance to sensitive variables. By analogy, we adapted invariant feature learning for domain generalization so as to address the domain-class dependency problem.

Domain generalization has been attracting considerable attention in recent years (Blanchard et al. (2011); Muandet et al. (2013); Shankar et al. (2018)). Note that it is different from domain adaptation in that we cannot obtain input and label data from target domain(s). Although the efficacy of domain-invariance-based methods had been known, Li et al. (2017) showed that non end-to-end methods such as DICA (Muandet et al. (2013)) and MTAE (Ghifary et al. (2015)) do not tend to outperform even vanilla CNN. Thus, end-to-end methods are desirable and can be divided into two categories: adversarial-learning-based methods such as DAN (Ganin et al. (2016); Xie et al. (2017)) and pre-defined-metric-based methods (e.g., Ghifary et al. (2017); Li et al. (2018b)).

In particular, IFLOC closely relates to DAN. Although DAN was originally invented for domain adaptation, Xie et al. (2017) showed its efficacy in domain generalization. Also, Xie et al. (2017) provided the intuitive explanation of the trade-off between classification accuracy and domain invariance. However, they did not provide any way to deal with the problem because its focus is invariant feature learning itself. Louppe et al. (2017) provided the similar analysis with Xie et al. (2017), but differs in that they focused on the relation between nuisance parameters (domains) and output distribution of a domain classifier. IFLOC also relates to *domain confusion loss* (Tzeng et al. (2015)) in that their encoders attempt to directly minimize Kullback-Leibler divergence (KLD) between output distribution of the discriminators and some domain distribution ($p(d|y)$ in IFLOC and uniform distribution in domain confusion loss), rather than deceive the discriminator as DAN.

There are several studies that address domain generalization without utilizing invariant feature learning. For example, Motiian et al. (2017); Li et al. (2018c) proposed to make use of semantic alignment, which attempts to make latent representation given class label ($p(h|y)$) identical within source domains. This approach was originally proposed in Gong et al. (2016) in domain adaptation context, but its efficacy for domain-class dependency is not obvious because it focuses on conditional probability shift. CrossGrad (Shankar et al. (2018)) is one of the recent state-of-the-art domain generalization methods, which utilizes data augmentation with adversarial examples. However, since the method relies on the assumption that $y$ and $d$ are independent, it might not be directly applicable to our setting. MLDG (Li et al. (2018a)), also one of the state-of-the-art methods, utilizes meta-learning. Since it makes no assumption about the relation between $y$ and $d$, it could be combined with our proposed method, though we have not experimentally confirmed it.

There are several kinds of distributional shifts other than domain-class dependency, such as conditional probability shift. Although the distinction between that shift and domain-class dependency is important, it has been received less attention. For example, Li et al. (2018c) claimed that conditional probability shift might harm the performance of domain-invariance-based methods, but our analysis in Sec.4.1.1 suggests that the root cause of the performance degradation is not it but domain-class dependency. They also proposed to correct the shift of $p(y)$ across source domains by aligning sampling frequency of each class across domains. However, this approach is not applicable to when some classes are rarely or never appear in some domains (e.g., as in WISDM dataset). In domain adaptation, Zhang et al. (2013); Gong et al. (2016) address the situation where $p(y)$ changes across source and target domains by estimating $p(y)$ change using unlabeled target data. However, this approach is not applicable (or necessary) to domain generalization because our problem setting is different from theirs in that we are agnostic on target domain and aim to care about $p(y)$ change within source domains instead.

# 3 PRELIMINARIES

## 3.1 PROBLEM STATEMENT OF DOMAIN GENERALIZATION

Denote $\mathcal{X}, \mathcal{Y}$, and $\mathcal{D}$ as the input feature, label, and domain spaces, respectively. With random variables $x \in \mathcal{X}, \ y \in \mathcal{Y}$, and $d \in \mathcal{D}$, we can define the probability distribution for each domain $d$ as $p(x, y|d)$. Here, we assume that $y$ and $d$ are discrete variables for simplicity. In domain generalization, we are given a training dataset consisting of $D_s = \{x_i^s, y_i^s\}_{i=1}^{n^s}$ for all $s \in \{1, 2, ..., m\}$. Here,

each $D_s$ corresponds to samples drawn from the source domain $p(x, y|d = s)$. Using the training dataset, we train a classifier $f : \mathcal{X} \rightarrow \mathcal{Y}$, and use the classifier to predict labels of samples drawn from the unknown target domain $p(x, y|d = t)$.

### 3.2 DOMAIN ADVERSARIAL NETWORKS FOR DOMAIN GENERALIZATION

In this section, we give a brief overview of DAN (Ganin et al. (2016)) given that our proposed method is an extension of it. DAN trains a domain discriminator that attempts to predict domains from latent representations encoded by an encoder, while simultaneously trains the encoder to remove domain information by deceiving the discriminator. This procedure ensures that there is no or little domain information in the representations, so a label classifier attached to the encoder can make robust predictions regarding unseen target domains.

Formally, we denote $f_E(x), q_M(y|h)$, and $q_D(d|h)$ ($E, M$, and $D$ are the parameters) as deterministic encoder, probabilistic model of label classifier, and that of domain discriminator, respectively. Then, the objective function of DAN is described as follows:

$$\min_{E,M} \max_D J(E, M, D) = \mathbb{E}_{x,d,y \sim p(x,d,y)}[\gamma \log q_D(d|h = f_E(x)) - \log q_M(y|h = f_E(x))] \quad (1)$$

Here, the second term in Eq.1 simply maximizes the log likelihood of $q_M$ as well as in standard classification problems. On the other hand, the first term corresponds to a minimax game between the encoder and discriminator, where the decoder $q_D(d|h)$ tries to predict $d$ from $h$ and the encoder $f_E(x)$ tries to fool $q_D(d|h)$.

As Xie et al. (2017) originally showed, the minmax game ensures that the learned representation has no or little domain information, i.e., the representation becomes domain-invariant. Such invariance makes a prediction from $h$ to $y$ independent from $d$, and therefore hopefully helps to build a classifier that correctly handle samples drawn from unknown domains. Below is a brief explanation.

Since $h$ is a deterministic mapping of $x$, the joint probability distribution of $h, d$ and $y$ can be defined as follows:

$$\tilde{p}_E(h, d, y) = \int_x \tilde{p}_E(x, d, h, y)dx$$
$$= \int_x p(x, d, y)\delta(f_E(x) = h)dx \quad (2)$$

Here, we use the notation of $\tilde{p}_E$ for the true probability distribution that depends on the encoder's parameter $E$. Using Eq.2, Eq.1 can be replaced as follows:

$$\min_{E,M} \max_D J(E, M, D) = \mathbb{E}_{h,d,y \sim \tilde{p}_E(h,d,y)}[\gamma \log q_D(d|h) - \log q_M(y|h)] \quad (3)$$

Assuming $E$ is fixed, the solutions $M^*$ and $D^*$ to Eq.3 obviously satisfy $q_{M^*}(y|h) = \tilde{p}_E(y|h)$ and $q_{D^*}(d|h) = \tilde{p}_E(d|h)$. Then, substituting $q_{M^*}$ and $q_{D^*}$ into Eq.3, we can obtain the following optimization problem depending only on $E$:

$$\min_E J(E) = -\gamma H_{\tilde{p}_E}(d|h) + H_{\tilde{p}_E}(y|h) \quad (4)$$

Solving Eq.4, we can obtain the solutions $M^*$, $D^*$, and $E^*$, which are in Nash equilibrium. Here, $H_{\tilde{p}_E}(d|h)$ means conditional entropy with joint probability distribution $\tilde{p}_E(d, h)$. Thus, minimizing the second term in Eq.4 intuitively means learning (the mapping function $f_E$ to) the latent representation $h$ which contains as much information about $y$ as possible. On the other hand, the first term can be regarded as a regularizer that attempts to learn $h$ which is invariant to $d$.

## 4 OUR APPROACH

### 4.1 ANALYSIS OF DOMAIN-CLASS DEPENDENCY

We address domain generalization under domain-class dependency, i.e., the situation where $p(y|d) \neq p(y)$ holds. Although the issue had been overlooked, it is common in real-world datasets given that they can have the dependency in nature, e.g., nocturnal annimals (class) do not tend to

appear in daylight (domain), and the dependency in such datasets is often not corrected unlike in standard benchmark datasets. To address the problem, we expand the analysis of Xie et al. (2017) to theoretically show that domain-class dependency causes the trade-off between accuracy and invariance, and to consider to what extent the latent representation should become invariant.

### 4.1.1 TRADE-OFF CAUSED BY DOMAIN-CLASS DEPENDENCY

We first show that the performance of DAN explained in the previous section suffers from the existence of domain-class dependency. The following analysis also suggests that all of the methods that utilize domain-invariant representation suffer from the dependency. Concretely, we show that the domain-class dependency causes the trade-off between classification accuracy and domain invariance: when $d$ and $y$ are not statistically independent, any $E$ cannot optimize the first and second term in Eq.4 at the same time. In this analysis, for simplicity, we assume that we can obtain any $\tilde{p}_E(y|h), \tilde{p}_E(d|h)$, i.e., the models have enough capacity and there are no optimization difficulties.

To begin with, we consider only the first term in Eq.4 and address the optimization problem:

$$\min_E J_1(E) = -\gamma H_{\tilde{p}_E}(d|h) \tag{5}$$

Using the property of entropy, $H_{\tilde{p}_E}(d|h)$ is bounded as follows:

$$H_{\tilde{p}_E}(d|h) \leq H(d) \tag{6}$$

Here, $H_{\tilde{p}_E}(d|h) = H(d)$ holds only if $h$ and $d$ are independent. Thus, Eq.5 has the solution $E^{1*}$, which satisfies the following condition:

$$H_{\tilde{p}_{E^{1*}}}(d|h) = H(d) \tag{7}$$

Eq.7 suggests that the regularizer in DAN is intended to remove all information about domains from latent variables, thereby making domains and latent variables independent.

Next, we analogically consider only the second term in Eq.4, thereby addressing the following optimization problem:

$$\min_E J_2(E) = H_{\tilde{p}_E}(y|h) \tag{8}$$

Since conditional entropy $H(a|b)$ has a minimum value when $b$ contains all information about $a$, Eq.8 has the solution $E^{2*}$, which satisfies the following equation:

$$H_{\tilde{p}_{E^{2*}}}(d|h) = H_{\tilde{p}_{E^{2*}}}(d|h, y) \tag{9}$$

Using Eq.9 and the property of entropy: $H(a|b, c) \leq H(a|b)$, we can obtain the following condition:

$$H_{\tilde{p}_{E^{2*}}}(d|h) = H_{\tilde{p}_{E^{2*}}}(d|h, y) \leq H(d|y) \tag{10}$$

Eq.10 implies that $h$ has at least as much information about $d$ as $y$ does. Now, we assume that $y$ and $d$ are not independent, i.e., domain-class dependency exists, and obtain the following condition:

$$H_{\tilde{p}_{E^{2*}}}(d|h) \leq H(d|y) < H(d) \tag{11}$$

Considering Eq.7 and Eq.11, $E^{1*} \neq E^{2*}$ holds. This means that when $y$ and $d$ are not independent, there is no solution $E$ that optimizes Eq.5 and Eq.8 at the same time, i.e., there is a trade-off between classification accuracy and domain invariance.

It is worth noting that although Li et al. (2018c) claimed that conditional probability shift (the causal structure $y \rightarrow x$) could harm the domain generalization performance of invariance-based methods, this analysis suggests that it does not harm DAN as long as domain and class are independent. It can be confirmed by considering Eq.7 and Eq.10; even when the shift occurs, i.e., $H(y|x, d) < H(y|x)$ holds and then $H_{\tilde{p}_E}(y|h, d) \leq H_{\tilde{p}_E}(y|h)$ holds, it does not conflict with $H_{\tilde{p}_{E^*}}(d|h) = H(d|y) = H(d)$ as long as $H(d|y) = H(d)$ holds. In other words, we only need to infer latent variable $h$ that satisfies the causal structure $y \rightarrow h \rightarrow x$ to avoid the trade-off. Although Gong et al. (2016) showed the similar result in domain adaptation context, it has been overlooked in domain generalization.

### 4.1.2 Optimal Domain-Invariance under Domain-Class Dependency

If we cannot avoid the trade-off, the next question is how to deal with it, i.e., to what extent the representation should become domain-invariant for domain generalization tasks. We propose to maximize domain-invariance within a range that does not interfere with classification accuracy, rather than merely enforcing domain-invariance without any constraint. The reason for the constraint is that the primary purpose of domain generalization is classification for unseen domains rather than domain-invariance itself, and the improvement of the invariance could harm the classification performance for them. For example, in WISDM, if we know the target activity (class) was performed by not an old but yound man (domain), we can predict it was jogging with higher probability, so we should avoid removing such domain information that is useful in the classification task. As another example, if the target domain has the similar characteristics as a certain source domain (or as an extreme case, $p(x, y|d = s) = p(x, y|d = t)$ holds), giving priority to domain-invariance obviously interferes with the domain generalization performance.

Given that Eq.10 is the necessary condition where we can build an optimal classifier, we can write the optimization problem of maximizing domain-invariance within a range that does not interfere with classification accuracy as follows:

$$\min_E J(E) = -\gamma H_{\tilde{p}_E}(d|h) \tag{12}$$

$$subject\ to\ H_{\tilde{p}_E}(d|h) \leq H(d|y) \tag{13}$$

Continuing, we can obtain the solution $E^*$, which obviously satisfies $H_{\tilde{p}_{E^*}}(d|h) = H(d|y)$. More specifically, when we want to maximize domain-invariance (Eq.12) within the range that does not interfere with accuracy (Eq.13), the solution satisfies $H_{\tilde{p}_{E^*}}(d|h) = H(d|y)$. So *without interfering with classification accuracy we can remove domain information from $h$ to the extent that $H_{\tilde{p}_E}(d|h) = H(d|y)$ holds*, i.e., $h$ has as much information about $d$ as $y$ does.

### 4.2 Proposed Method

Based on the above analysis, the remaining challenge is how to impose such regularization that makes $H_{\tilde{p}_E}(d|h) = H(d|y)$ hold. Although DAN might be able to achive that condition by carefully tuning the regularizer ($\gamma$ in Eq.1), such tuning is time-consuming and impracticable as suggested in our experiments. Alternatively, we propose a novel method called IFLOC, modifying DAN's regularization term: while the encoder of DAN attempts to fool a discriminator, that of IFLOC attempts to directly minimize KLD between $p(d|y)$ and $q_D(d|h)$. Formally, IFLOC solves the following joint optimization problem by alternating gradient descent.

$$\min_{E,M} J(E, M) = \mathbb{E}_{x,d,y \sim p(x,d,y)}[\gamma D_{KL}[p(d|y)|q_D(d|h = f_E(x))] - \log q_M(y|h = f_E(x))] \tag{14}$$

$$\min_D J(E, D) = \mathbb{E}_{x,d \sim p(x,d)}[-\log q_D(d|h = f_E(x))] \tag{15}$$

The second term in Eq.14 and Eq.15 respectively means maximization of log-likelihood of $q_M$ and $q_D$ as well as DAN. However, the first term in Eq.14 differs from DAN in that it is intended to satisfy $q_D(d|h) = p(d|y)$ for almost every $(y, h)$ pair.

Next we show that the regularization of IFLOC is intended to achieve $H_{\tilde{p}_E}(d|h) = H(d|y)$. Similarly to Section 3.2, $D^*$ and $M^*$, which are the solutions to Eq.14 and Eq.15 with fixed $E$, obviously satisfy $q_D^* = \tilde{p}_E(d|h), q_M^* = \tilde{p}_E(y|h)$. Thus Eq.14 can be written as follows:

$$\min_E J(E) = \mathbb{E}_{h,y \sim \tilde{p}_E(h,y)}[\gamma D_{KL}[p(d|y)|\tilde{p}_E(d|h)]] + H_{\tilde{p}_E}(y|h) \tag{16}$$

Since the minimization of the KLD term does not interfere with the second term optimization, $E^*$, which is the solution to Eq.16 and in Nash equilibrium, satisfies $\mathbb{E}_{h,y \sim \tilde{p}_{E^*}(h,y)}[D_{KL}[p(d|y)|\tilde{p}_{E^*}(d|h)]] = 0$. Then, $H_{\tilde{p}_{E^*}}(d|h) = H(d|y)$ obviously holds.

Note that we cannot obtain true $p(d|y)$, but we can use a maximum likelihood or maximum a posteriori estimator for it. Also, we could use some divergences other than $D_{KL}[p(d|y)|q_D(d|h)]$ in Eq.14, e.g., $D_{KL}[q_D(d|h)|p(d|y)]$, but in doing so, we could not observe performance gain, so we discontinued testing them.

Table 1: Sample sizes for each domain-class pair in BMNISTR. Those for the classes 0∼4 are variable across domains, whereas the classes 5∼9 have identical sample sizes across domains.

| Dataset | Class | M0 | M15 | M30 | M45 | M60 | M75 |
|---------|-------|-----|-----|-----|-----|-----|-----|
| BMNISTR-1 | 0∼4 | 100 | 85 | 70 | 55 | 40 | 25 |
| | 5∼9 | 100 | 100 | 100 | 100 | 100 | 100 |
| BMNISTR-2 | 0∼4 | 100 | 80 | 60 | 40 | 20 | 0 |
| | 5∼9 | 100 | 100 | 100 | 100 | 100 | 100 |
| BMNISTR-3 | 0∼4 | 100 | 90 | 80 | 70 | 60 | 50 |
| | 5∼9 | 100 | 100 | 100 | 100 | 100 | 100 |
| BMNISTR-4 | 0∼4 | 100 | 25 | 100 | 25 | 100 | 25 |
| | 5∼9 | 100 | 100 | 100 | 100 | 100 | 100 |

## 5 EXPERIMENTS

### 5.1 DATASETS

**BMNISTR** We created the Biased Rotated MNIST dataset (BMNISTR) by modifying the sample size of MNISTR (Ghifary et al. (2015)) so that class distribution differs among the domains. Specifically, we created four variants of MNISTR that have different types of domain-class dependency, referred to as BMNISTR-1 through BMNISTR-4. As shown in Table 1, BMNISTR-1, -2, and -3 have similar trends but different degrees of dependency; BMNISTR-1 and BMNISTR-4 differ in trends. In MNISTR, each class is represented by 10 digits. Each domain was created by rotating images by 15 degree increments: 0, 15, 30, 45, 60, and 75 (referred to as M0, ..., M75). Each image is cropped to 16 x 16 in accordance with Ghifary et al. (2015). In training, we employed one-domain-leave-out setting: trained on five of the six domains and then tested using the remaining one. We used two convolution layers and two fully-connected (FC) layers (with nonlinear activations) as the encoder, three FC layers as the classifier, and two FC layers as the discriminator.

**PACS** The PACS dataset (Li et al. (2017)) has 9991 images across 7 categories (dog, elephant, giraffe, guitar, house, horse, and person) and 4 domains comprising different stylistic depictions (Photo, Art painting, Cartoon, and Sketch). It has domain-class dependency probably because samples in some <domain, class> pairs are difficult to obtain. For example, $p(y = \text{person}|d = \text{Phot})$ is much higher than $p(y = \text{person}|d = \text{Sketch})$, which indicates that photos of person are easier to obtain than those of animals, but sketches of persons are more difficult to obtain than those of animals in the wild. The concrete sample sizes for each category and style is shown in Table 4 in appendix. In training, we employed one-domain-leave-out setting as well as in BMNISTR, and used the ImageNet pre-trained AlexNet CNN (Krizhevsky et al. (2012)) as the base network, following previous studies (Li et al. (2017; 2018a)). The two-FC-layer discriminator was connected to the last FC layer, following Ganin et al. (2016).

**WISDM** The WISDM Activity Prediction dataset contains sensor data of accelerometers for six human activities (walking, jogging, upstairs, downstairs, sitting, and standing) performed by 36 users (domains). WISDM suffers from the dependency due to the reason noted in Sec.1. The concrete sample sizes for each user and activity is shown in Table 5 in appendix. Referring to Andrey (2017), we use the sliding-window procedure with 60 frames (=3 seconds) and 20-frame overlap. The total number of samples was 54455. In training, we used randomly chosen <10 / 26>, <16 / 20>, and <26 / 10> users as <source / target> domains. We parameterized the encoder using three convolution layers followed by one FC layer and the classifier by logistic regression, following previous studies (Yang et al. (2015); Iwasawa et al. (2017)). The two-FC-layer discriminator was connected to the output of the encoder.

### 5.2 BASELINES

To demonstrate the efficacy of the proposed method IFLOC, we compared it with the following methods. **(1) CNN** is a vanilla convolutional networks trained on the aggregation of data from all source domains. Although CNN has no special treatments for domain generalization, Li et al. (2017) reports that it outperforms many traditional domain generalization methods. **(2) DAN (Xie et al.**

Table 2: Mean F scores for the classes 0∼4 and classes 5∼9 with the target domain M0.

| Dataset | Class | CNN | DAN | IFLOC-Abl | IFLOC | Relative Improvement of IFLOC to IFLOC-Abl |
|---|---|---|---|---|---|---|
| BMNISTR-1 | 0∼4 | 83.86 | 84.54 | 87.46 | **90.62** | 3.6% |
|  | 5∼9 | 83.90 | 85.24 | 86.46 | **88.10** | 1.9% |
| BMNISTR-2 | 0∼4 | 84.76 | 86.20 | 86.42 | **89.58** | 3.7% |
|  | 5∼9 | 83.36 | 85.22 | 85.62 | **86.86** | 1.4% |
| BMNISTR-3 | 0∼4 | 82.54 | 85.30 | 88.60 | **89.64** | 1.2% |
|  | 5∼9 | 82.18 | 85.80 | 87.60 | **89.04** | 1.6% |
| BMNISTR-4 | 0∼4 | 71.26 | 79.22 | 76.56 | **80.02** | 4.5% |
|  | 5∼9 | 78.62 | **83.14** | 82.94 | 82.80 | -0.2% |

**(2017))** is expected to generalize across domains via invariant feature learning, but it has the trade-off between domain invariance and classification accuracy as explained in Section 4.1.1. We trained DAN with a gradient reverse layer following Ganin et al. (2016); Xie et al. (2017). Also, we used **(3) IFLOC-Abl**, which is a version of IFLOC modified for ablation studies. IFLOC-Abl replaces $D_{KL}[p(d|y)|\tilde{p}_E(d|h)]$ in Eq.14 of $D_{KL}[p(d)|\tilde{p}_E(d|h)]$, so it attempts to learn the representation that is completely invariant to domains or make $H(d|h) = H(d)$ hold as well as DAN. Comparing IFLOC and IFLOC-Abl, we measured the genuine effect of taking domain-class dependency into account. In training IFLOC and IFLOC-Abl, we cannot obtain true $p(d|y)$ and $p(d)$, so we used maximum likelihood estimators of them for calculating the KLD terms.

## 5.3 EXPERIMENTAL SETTINGS

For all the datasets and methods, we used RMSprop for optimization. And we set the learning rate, batch size, and the number of iterations as 5e-4, 128, and 10k for BMNISTR; 5e-5, 64, and 10k for PACS; 1e-4, 128, and 30k for WISDM, respectively. For DAN, IFLOC-Abl, and IFLOC we optimized the weighting parameter $\gamma$ from $\{0.0001, 0.001, 0.01, 0.1, 1, 10\}$, and used the $\gamma$ annealing following Ganin et al. (2016). In all the experiments, we split source data into 80% of training data and 20% of validation data, assuming that target data are not absolutely available in the training phase. We conducted experiments multiple times with different seeds. Specifically, we trained on 10 and 25 seeds in BMNISTR and WISDM, chose the best hyperparameter that achieved the highest validation accuracies measured in each epoch, and reported the mean scores (accuracies and f-values) for the hyperparameter. In PACS, since it requires a long time to train on, we chose the best $\gamma$ from $\{0.0001, 0.001, 0.01, 0.1\}$ with three experiment, and reported the mean scores in experiments with 20 seeds in total. Also, we empirically measured the level of domain-invariance by training a post-hoc classifier that is intended to predict $d$ over learned representation, following previous studies (Xie et al. (2017); Iwasawa et al. (2017); Moyer et al. (2018)). Specifically, we trained the classifier with 400 hidden units on 10k iterations (by RMSprop optimizer with a 0.001 learning rate and 128 batch size) with the data that is used for training the models. We then evaluated the domain classification accuracy (referred to as D-Acc) 10 times at equal intervals during training, and reported D-Acc in the nearest time when the validation accuracy is maximized.

## 5.4 RESULTS

We first investigated how domain-class dependency affects the performance of domain-invariance-based methods. In Table 2, we compared mean f-scores for the classes 0 through 4 and classes 5 through 9 in BMNISTR with the target domain M0. Recall that sample sizes for the classes 0∼4 are variable across domains, whereas the classes 5∼9 has identical sample sizes across domains (Table 1). The f-scores show that IFLOC outperformed DAN and IFLOC-Abl in most dataset-class pairs, which supports that domain-class dependency depresses the performance of domain-invariance-based methods and that IFLOC can mitigate the problem. Futher, relative improvement of IFLOC to IFLOC-Abl is more significant for the classes 0∼4 than 5∼9 in BMNISTR-1, BMNISTR-2, and BMNISTR-4, suggesting that IFLOC tends to increase performance more significantly for classes where the domain-class dependency occurs. Also, the improvement is more significant in

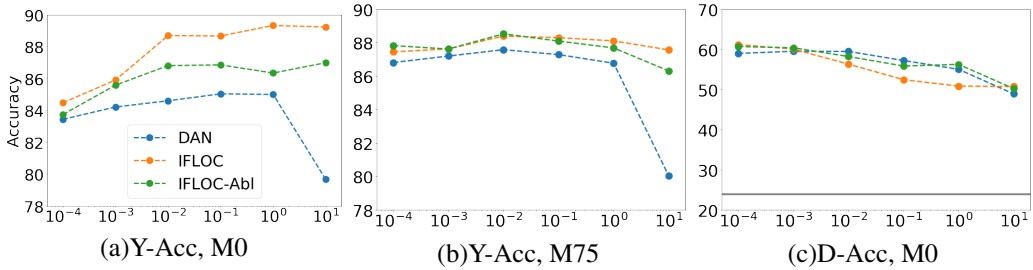

Figure 2: Class accuracy (Y-Acc) and domain accuracy (D-Acc) with various $\gamma$ in BMNISTR-1. Each caption shows the metric name (Y-Acc or D-Acc) and target domain.

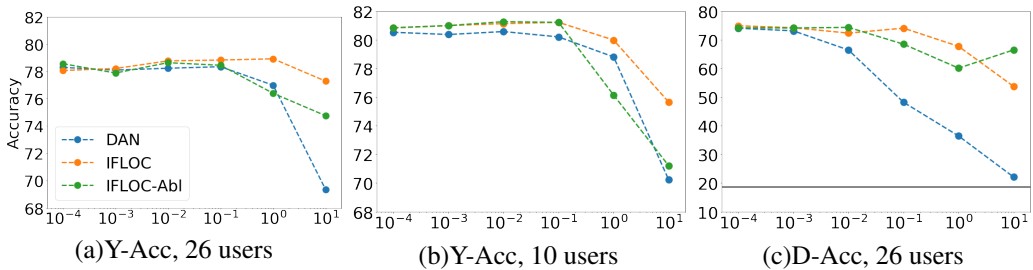

Figure 3: Class accuracy (Y-Acc) and domain accuracy (D-Acc) with various $\gamma$ in WISDM. Each caption shows the metric name (Y-Acc or D-Acc) and target domain.

BMNISTR-1 than in BMNISTR-3, suggesting that the stronger the domain-class dependency is, the lower the performance of domain-invariance-based methods becomes. Finally, although the dependencies of BMNISTR-1 and BMNISTR-4 have different trends as described in Table 1, IFLOC improved f-scores in both datasets.

Next we investigated the relationship between the strength of regularization and performance. Figures 2 and 3 show the hyperparameter sensitivity of class accuracies (Y-Acc) and domain accuracies (D-Acc) for DAN, IFLOC-Abl, and IFLOC. Note that the gray line in Figures 2-(c) and 3-(c) shows the trivial baseline predicting the majority label. From these figures, we can make the following observations. **(1)** All the methods including IFLOC could improve the invariance by using stronger regularizer. Concretely, Figures 2-(c) and 3-(c) show D-Acc tends to become low (invariance becomes high) for all the models when the regularizer becomes strong (such as $\gamma = 1$ or 10) except that IFLOC-Abl has high D-Acc with $\gamma = 10$ in Figure 3-(c). That high D-Acc might be because the validation accuracy achieved the highest value before the domain-invariance matured. (Recall that the more the representation becomes invariant, the lower the accurary becomes under the trade-off). **(2)** The training of IFLOC tends to be more stable than that of DAN when the regularizer becomes strong. Figures 2-(a,b) and 3-(a,b) show that IFLOC and IFLOC-Abl could achieve higher Y-Acc than DAN when $\gamma = 1$ or 10, i.e., the regularization is strong, except for IFLOC-Abl with $\gamma = 1$ in Figure 3-(b). This tendency might be because the regularizer of IFLOC is KLD and thus bounded by 0, in contrast to that of DAN that can increase to infinity and destabilize the traininig. **(3)** IFLOC, as it was designed, does not tend to decrease classification accuracy with strong regularizer, and thus IFLOC is robust toward hyperparameter choice. Figures 2-(b) and 3-(a,b) show that while Y-Acc of IFLOC-Abl decreases with strong regularization (such as when $\gamma = 1$ or 10), that of IFLOC does not decrease as much.

Finally, we compared mean accuracies (with standard errors) in both synthetic (BMNISTR) and standard benchmark (PACS and WISDM) datasets (Table 3). Note that the $H(d)/H(d|y)$ column is estimated from source data, which indicates the strength of domain-class dependency. IFLOC outperformed IFLOC-Abl in BMNISTR with all the target domains; PACS with photo, art_painting, and sketch target domains; and WISDM with 26- and 20-target-user domains. Also, IFLOC outperformed DAN in BMNISTR with all the target domains; PACS with photo and art_painting target

Table 3: Accuracies for each dataset and target domain

| Dataset | Target | $H(d)/H(d\|y)$ (%) | CNN | DAN | IFLOC-Abl | IFLOC |
|---------|--------|------------|-----|-----|-----------|-------|
| BMNISTR-1 | M0 | 101.2 | 83.9 ± 0.4 | 85.0 ± 0.4 | 87.0 ± 0.4 | **89.3 ± 0.4** |
| | M15 | 101.5 | 98.5 ± 0.2 | 98.5 ± 0.1 | 98.3 ± 0.2 | **98.8 ± 0.1** |
| | M30 | 101.6 | 97.5 ± 0.1 | 97.4 ± 0.1 | 97.6 ± 0.1 | **98.3 ± 0.2** |
| | M45 | 101.6 | 89.9 ± 0.9 | 90.2 ± 0.6 | 92.8 ± 0.5 | **93.3 ± 0.6** |
| | M60 | 101.3 | 96.7 ± 0.3 | 97.0 ± 0.2 | 96.6 ± 0.2 | **97.4 ± 0.2** |
| | M75 | 100.7 | 87.1 ± 0.5 | 87.3 ± 0.4 | 87.7 ± 0.5 | **88.1 ± 0.4** |
| | Avg | | 92.3 | 92.6 | 93.3 | **94.2** |
| BMNISTR-2 | Avg | | 92.3 | 92.2 | 93 | **94.2** |
| BMNISTR-3 | Avg | | 92.2 | 92.7 | 94 | **94.5** |
| BMNISTR-4 | Avg | | 90.6 | 91.7 | 91.6 | **92.9** |
| PACS | photo | 107.2 | 80.6 ± 0.3 | 81.1 ± 0.3 | 81.6 ± 0.3 | **82.9 ± 0.2** |
| | art_painting | 108.5 | 59.2 ± 0.4 | 60.1 ± 0.3 | 60.5 ± 0.4 | **61.2 ± 0.2** |
| | cartoon | 109.7 | 63.2 ± 0.3 | 64.3 ± 0.3 | **64.4 ± 0.4** | 63.8 ± 0.3 |
| | sketch | 101.5 | 58.2 ± 0.5 | 58.9 ± 0.4 | 58.1 ± 0.6 | **59.0 ± 0.5** |
| | Avg | | 65.3 | 66.1 | 66.2 | **66.7** |
| WISDM | 26 users | 107.1 | 78.3 ± 0.3 | 78.2 ± 0.3 | 78.4 ± 0.2 | **78.9 ± 0.3** |
| | 20 users | 104.2 | 79.7 ± 0.2 | **80.2 ± 0.3** | 79.7 ± 0.3 | 80.0 ± 0.3 |
| | 10 users | 103.5 | 80.6 ± 0.2 | 80.6 ± 0.2 | **81.2 ± 0.3** | **81.2 ± 0.3** |

domains; and WISDM with 26- and 10-target-user domains. This supports the importance of considering domain-class dependency in real-world datasets and the efficacy of the proposed model.

Table 3 also shows that when the number of source domains increased from 10 to 26 in WISDM, the improvement of IFLOC from IFLOC-Abl became insignificant. One possible reason is that WISDM with 10 target users has low domain-class dependency than with 26 target users as shown in the $H(d)/H(d|y)$ column. Another possible reason is the optimization difficulty. As Moyer et al. (2018) reported, in adversarial invariant feature learning, an encoder often overfits to the discriminator trained alongside that encoder, and does not provide truly invariant representation (the same problem can be observed in Figures 2-(c) and 3-(c)). We suspect that when the number of source domains increases, the optimization of the domain discriminator becomes difficult, which makes the encoder overfit to that poor discriminator and worsen the problem. Also, the improvement of IFLOC from IFLOC-Abl is less significant in WISDM than that in BMNISTR and PACS, which could be related to the same problem since the number of source domains for BMNISTR and PACS is smaller than that for WISDM. If the optimization difficulty prevents IFLOC from working properly, we might be able to mitigate it by using ideas from the studies that investigate the convergence and optimization difficulty in adversarial training (e.g., Nagarajan & Kolter (2017); Heusel et al. (2017); Balduzzi et al. (2018)).

# 6 CONCLUSION

In this paper, we addressed domain generalization under *domain-class dependency*, which was overlooked by most prior domain generalization methods relying on domain-invariant representation. We theoretically showed the importance of considering the dependency and the way to overcome the problem by expanding the analysis of Xie et al. (2017). We then proposed a novel method IFLOC, which maximizes domain-invariance within a range that does not interfere with classification accuracy. Empirical validations show the superior performance of IFLOC to the baseline methods, supporting the importance of the domain-class dependency in domain generalization tasks and the efficacy of the proposed method for overcoming the issue. Future work includes applying the regularization idea of making $H(d|h) = H(d|y)$ to other methods, e.g., Conditional VAE (Louizos et al. (2016)) or CrossGrad (Shankar et al. (2018)) because they have clear and tractable data generating process but assume the independence of $y$ and $d$. Also intended is to use it for transfer learning tasks in a few-shot setting (e.g., life-long learning) where domain-class dependency is likely to occur due to scarce sample size.

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

# A DOMAIN-CLASS DEPENDENCY IN PACS AND WISDM

Table 4: Sample sizes for each <domain, class> pair in PACS dataset. The column shows category name while and index shows style.

|  | Guitar | House | Giraffe | Person | Horse | Dog | Elephant |
|---|---|---|---|---|---|---|---|
| Art Painting | 184 | 295 | 285 | 449 | 201 | 379 | 255 |
| Cartoon | 135 | 288 | 346 | 405 | 324 | 389 | 457 |
| Photo | 186 | 280 | 182 | 432 | 199 | 189 | 202 |
| Sketch | 608 | 80 | 753 | 160 | 816 | 772 | 740 |

Table 5: Sample sizes for each <domain, class> pair in WISDM dataset. The column shows activity name and the index shows user id.

|  | Jogging | Walking | Upstairs | Downstairs | Sitting | Standing |
|---|---|---|---|---|---|---|
| User 1 | 145 | 742 | 108 | 224 | 160 | 78 |
| User 2 | 142 | 481 | 282 | 186 | 0 | 0 |
| User 3 | 645 | 654 | 240 | 231 | 780 | 267 |
| User 4 | 637 | 619 | 237 | 214 | 113 | 78 |
| User 5 | 614 | 650 | 229 | 210 | 56 | 80 |
| User 6 | 637 | 574 | 101 | 86 | 0 | 0 |
| User 7 | 588 | 617 | 81 | 69 | 81 | 33 |
| User 8 | 599 | 621 | 160 | 171 | 102 | 79 |
| User 9 | 599 | 308 | 269 | 206 | 123 | 94 |
| User 10 | 597 | 625 | 119 | 118 | 71 | 95 |
| User 11 | 610 | 616 | 188 | 115 | 150 | 81 |
| User 12 | 626 | 356 | 0 | 0 | 77 | 51 |
| User 13 | 620 | 604 | 217 | 131 | 0 | 0 |
| User 14 | 0 | 624 | 68 | 76 | 147 | 96 |
| User 15 | 318 | 610 | 167 | 162 | 81 | 73 |
| User 16 | 602 | 650 | 212 | 187 | 0 | 81 |
| User 17 | 0 | 706 | 142 | 147 | 0 | 63 |
| User 18 | 593 | 658 | 178 | 189 | 0 | 0 |
| User 19 | 661 | 690 | 406 | 141 | 0 | 0 |
| User 20 | 611 | 310 | 149 | 144 | 32 | 25 |
| User 21 | 616 | 537 | 130 | 141 | 112 | 81 |
| User 22 | 613 | 327 | 239 | 94 | 0 | 0 |
| User 23 | 42 | 301 | 66 | 86 | 60 | 0 |
| User 24 | 0 | 626 | 209 | 191 | 75 | 152 |
| User 25 | 641 | 666 | 194 | 140 | 76 | 65 |
| User 26 | 513 | 853 | 220 | 165 | 132 | 161 |
| User 27 | 701 | 841 | 231 | 192 | 105 | 128 |
| User 28 | 477 | 622 | 240 | 199 | 78 | 140 |
| User 29 | 548 | 646 | 168 | 164 | 78 | 139 |
| User 30 | 309 | 349 | 269 | 179 | 0 | 0 |
| User 31 | 550 | 641 | 154 | 145 | 0 | 0 |
| User 32 | 0 | 644 | 0 | 0 | 0 | 0 |
| User 33 | 322 | 346 | 0 | 0 | 0 | 0 |
| User 34 | 587 | 584 | 0 | 0 | 0 | 0 |
| User 35 | 457 | 549 | 178 | 110 | 124 | 116 |
| User 36 | 808 | 879 | 212 | 128 | 124 | 104 |

