# OpenReview forum: "Domain Generalization via Invariant Representation under Domain-Class Dependency"
_ICLR.cc/2019/Conference_

### Official Review · AnonReviewer3 · 2018-11-02
**The proposed problem seems similar to traditional conditional distribution matching problem.**

**Rating:** 5
**Confidence:** 4

**Review:**

The paper proposed a problem that most prior methods overlooked the underlying dependency of classes on domains, namely p (y|d) \= p(y).   Figure 1 is used to illustrate this issue.

If the conditional probability of source domain and target domain is not equal (i.e., p(y|x_S) \= p(y|x_T)  ), the optimal invariance can lead the same generalization problem.   Unfortunately, a lot of works has been done [1,2] in matching domain classifier or conditional probability.  It is desirable to discuss the difference between these two problems and compared with the missing references in experiments.

It is also suggested to conduct the analysis of why the datasets satisfy the assumption of the dependence of class and domains.

Reference:
[1] Flexible Transfer Learning under Support and Model Shift, NIPS 2014.
[2]Conditional Adversarial Domain Adaptation, NIPS 2018

---

> ### Author Response · Authors · 2018-11-14
> **Response to Reviewer 3 [1/4]**
>
> Thank you for your critical feedback.
> We hope to clarify and address your concerns and questions.
> We respond in detail to each comment below.
>
>
> ### Reply to “it is desirable to discuss the difference between these two problems”
>
> In our understanding, your main concern is the novelty of our problem setting, i.e., “is domain generalization under domain-class dependency (p(y|d) \neq p(y)) is different from domain adaptation under p(y|x_S) \neq p(y|x_T) ?”
> We acknowledge that we lack the discussion about the difference between these two (though they are indeed considerably different problems), so we have added the below discussion to the paper and emphasized the novelty of our problem setting.
>
> Firstly, the paper addresses {\em domain generalization}, not domain adaptation, as noted in abstract, Sec.1, etc.
> These two have different assumptions and purposes.
> Concretely, domain adaptation methods require either labeled or unlabeled data from the target domain at training time.
> In contrast, domain generalization methods do not require any data from target domains during training but instead, require labeled data from several source domains.
> Then the methods collectively exploit them so that the trained system can handle new domains without any adaptation step.
>
> Due to the difference, domain adaptation methods are not always applicable to domain generalization.
> For example, Wang+2014, which you suggested for us, transform unlabeled target data so that they can correct distributional shift, but in domain generalization, target data are unavailable.
> Also, please note that we care about the shifts within source domains, because domain generalization methods are agnostic on the target domain.
> So we think p(y|x_S) \neq p(y|x_T) should be rewritten as p(y|x, d) \neq p(y|x) (we call it conditional probability shift) in domain generalization, so that clarify we focus on the shift within source domains (not between S and T).
>
> Secondly, conditional probability shift (p(y|x, d) \neq p(y|x)), which is often caused by the causal structure y -> x, is not a sufficient condition for p(y|d) \neq p(y).
> While conditional probability shift was previously addressed by Li+2018 in domain generalization context, domain-class dependency has been overlooked.
> The relation between these two problems is illustrated in Figure 1 in our updated paper.
>
> Thirdly and most importantly, in domain generalization, conditional probability shift does not cause the trade-off problem as long as the domain-class dependency does not exist.
> In other words, p(y|x, d) \neq p(y|x) is not a root cause of the trade-off problem, but domain-class dependency is, so it is essential to consider and address domain-class dependency problem.
> Again the relation between these two problems is illustrated in Figure 1 in our updated paper.

---

> > ### Author Response · Authors · 2018-11-14
> > **Response to Reviewer 3 [2/4]**
> >
> >
> > Based on the above discussion (Response to Reviewer 3 [1/4]), we have updated the paper. Below are the details.
> >
> > 1. To make it clear that our purpose is improving {\em domain generalization performance}, we have changed the title of the paper to ”domain generalization via invariant representation under domain-class dependency.”
> >
> > 2. To make it clear that domain generalization differs from domain adaptation, We have added the following sentences:
> > ```
> > (Sec.2, para.2) Domain generalization has been attracting considerable attention in recent years (Blanchard et al. (2011); Muandet et al. (2013); Shankar et al. (2018)). Note that it is different from domain adaptation in that we cannot obtain input and label data from the target domain(s).
> > (Sec.2, para.5) In domain adaptation, Zhang et al. (2013); Gong et al. (2016) address the situation where p(y) changes across source and target domains by estimating p(y) change using unlabeled target data. However, this approach is not applicable (or necessary) to domain generalization because our problem setting is different from theirs in that we are agnostic on target domain and aim to care about p(y) change within source domains instead.
> > ```
> >
> > 3. To make it clear that conditional probability shift (p(y|x, d) \neq p(y|x)) and domain-class dependency (p(y|d) \neq p(y)) are different problems, and the latter is the root cause of the trade-off problem, we have added the following sentences:
> > ```
> > (Sec.1, para.3) We define domain-class dependency as the situation where domain and class labels are statistically dependent due to some common latent factor (z) of y and d (Figure 1-right).
> > (Sec.1, para.3) Domain-class dependency might be similar to the situation where p(y|x) and p(x) change across domains due to the causal structure y → x (Zhang et al. (2013); Gong et al. (2016) in domain adaptation and Li et al. (2018c) in domain generalization), which we call conditional probability shift. However, the shift does not cause the trade-off as long as y and d are independent (Figure 1-left), so it is necessary to focus on the relationship between y and d.
> > (Sec.2, para.5) There are several kinds of distributional shifts other than domain-class dependency, such as conditional probability shift. Although the distinction between that shift and domain-class dependency is important, it has been received less attention. For example, Li et al. (2018c) claimed that conditional probability shift might harm the performance of domain-invariance-based methods, but our analysis in Sec.4.1.1 suggests that the root cause of the performance degradation is not it but domain-class dependency.
> > ```
> >
> > 4. To show why conditional probability shift does not cause the trade-off problem and why domain-class dependency is important, we added the following sentence:
> > ```
> > (Sec.4.1.1, the last para.) It is worth noting that although Li et al. (2018c) claimed that conditional probability shift (the causal structure y → x) could harm the domain generalization performance of invariance-based methods, this analysis suggests that it does not harm DAN as long as domain and class are independent. It can be confirmed by considering Eq.7 and Eq.10; even when the shift occurs, i.e., H(y|x, d) < H(y|x) holds and then H(y|h, d) ≤ H(y|h) holds, it does not conflict with H(d|h) = H(d|y) = H(d) as long as H(d|y) = H(d) holds. In other words, we only need to infer latent variable h that satisfies the causal structure y → h → x to avoid the trade-off. Although Gong et al. (2016) showed a similar result in domain adaptation context, it has been overlooked in domain generalization.
> > ```

---

> > > ### Author Response · Authors · 2018-11-14
> > > **Response to Reviewer 3 [3/4]**
> > >
> > >
> > > ### Reply to “it is desirable to compare with the missing references in experiments”
> > >
> > > Thank you for introducing two important works.
> > > These works greatly help us re-consider the novelty of our work.
> > >
> > > However, currently, we do not plan to conduct comparative experiments with them for two reasons.
> > > Firstly, both Wang+2014 and Long+2018 address domain adaptation, so their methods are not directly applicable to our setting (domain generalization).
> > > Secondly, as we discussed in reply to comment 1, p(y|x, d) \neq p(y|x) is not a sufficient condition for p(y|d) \neq p(y), and this paper addresses the trade-off problem caused by the latter.
> > > More specifically, we acknowledge that comparing our method with methods invented for addressing p(y|x, d) \neq p(y|x) is very interesting research direction, but is out of scope of this paper, given our three contributions: the novel problem setting (domain generalization under domain-class dependency); the theoretical analysis which derives the novel approach to address the problem (to regularize latent representation so that H(d|y)=H(d|h) holds); and to confirm the efficacy of the proposed approach with the novel algorithm (IFLOC).
> > >
> > >
> > > Below are the detailed comments to each paper.
> > >
> > > Wang+2014: They consider domain adaptation under a general case where both the support and the model (p(y|x)) change across domains. They proposed to transform both X and Y by a location-scale shift to achieve transfer between domains. Our paper differs from theirs in that we focus on domain generalization (while they focus on domain adaptation), we focus on p(y|d) \neq p(y) (they focus on p(y|x, d) \neq p(y|x)), we focus on classification (they focus on regression), and we use DNN to extract latent features (they use location-scale shift to transform target data into source data).
> > >
> > > Long+2018: They address two issues in domain adaptation: (1) when data distributions embody complex multimode structures, adversarial domain adaptation may fall into mode collapse; (2) adversarial adaptation of a particular layer is not sufficient to bridge the domain shifts. To address these issues, they proposed to conditioning domain discriminator on the middle layer and output layer of the classifier. Our paper differs from theirs in that we focus on domain generalization (while they focus on domain adaptation), we focus on p(y|d) \neq p(y) (they do not mention the type of distributional shift), and we avoid removing all domain information from latent features.
> > > (Also, please note that the NIPS version of the paper is unavailable now.)

---

> > > > ### Author Response · Authors · 2018-11-14
> > > > **Response to Reviewer 3 [4/4]**
> > > >
> > > >
> > > > ### Reply to “It is also suggested to conduct the analysis of why the datasets satisfy the assumption of the dependence of class and domains”
> > > >
> > > > We agree that “the analysis of why the datasets satisfy the assumption of the dependence of class and domains” would improve the paper, and some analysis is already done in the paper.
> > > > So we would very appreciate if you could specify what kind of analysis would improve the paper more.
> > > > Below are the analyses we already did in the paper.
> > > >
> > > > 1. BMNISTR is the synthetic dataset which we created by modifying MNISTR. As noted in Sec.5.1., BMNISTRs have several types of domain-class dependency, and the reason for each type is noted in Sec.5.1. Also, the performances of IFLOC in each dependency type are discussed in the 1st-paragraph in Sec.5.4.
> > > >
> > > > 2. PACS has the dependency probably because samples in some <domain, class> pairs are difficult to obtain, as noted in Sec.5.1. The reasons why some <domain, class> pairs become difficult to obtain is discussed in Sec.1 (data characteristics and data-collection errors).
> > > >
> > > > 3. WISDM has the dependency due to the reason discussed in Sec.1.
> > > >
> > > >
> > > > Also, we have added the below discussion regarding the datasets.
> > > >
> > > > 1. We have added the table of the concrete sample sizes for each <domain, class> pair in PACS and WISDM dataset to Appendix.
> > > >
> > > > 2. The reason why WISDM has the dependency was already noted in Sec.1, but we have added it to Sec.5.1 again.
> > > >
> > > > 3. We have provided the concrete example of domain-class dependency in PACS as follows:
> > > > ```
> > > > (Sec.5.1, para.2) For example, p(y = person|d = Phot) is much higher than p(y = person|d = Sketch), which indicates that photos of person are easier to obtain than those of animals, but sketches of persons are more difficult to obtain than those of animals in the wild.
> > > > ```
> > > >
> > > > 4. We have cited Zhang+2013 to support the fact that that domain-class dependency often happens in real-world dataset as follows:
> > > > ```
> > > > (Sec.1, para.4) Unfortunately, domain-class dependency is common in real-world datasets as shown in Zhang et al. (2013).
> > > > ```
> > > >
> > > > ### References
> > > >
> > > > Kun Zhang, Bernhard Scholkopf, Krikamol Muandet, Zhikun Wang, “Domain Adaptation under Target and Conditional Shift”, ICML2013
> > > > Xuezhi Wang, Jeff Schneider, “Flexible Transfer Learning under Support and Model Shift”, NIPS2014
> > > > Mingsheng Long, Zhangjie Cao, Jianmin Wang, Michael I. Jordan, “Conditional Adversarial Domain Adaptation”, NIPS2018
> > > > Ya Li, Mingming Gong, Xinmei Tian, Tongliang Liu, Dacheng Tao, “Domain Generalization via Conditional Invariant Representations”, AAAI2018

---

### Official Review · AnonReviewer1 · 2018-11-02
**Simple and effective idea, good experiment design**

**Rating:** 7
**Confidence:** 5

**Review:**

This paper proposed to address domain generalization under inter-dependence of domains and classes. It motivates a new regularization term by analyzing an existing work, DAN. It shows that this term can improve the generalization performance when the classes and domains are not independent. Experiments are extensive and supportive.

I do not have many comments about this paper. It was a joy to read. The proposed idea is well motivated. It is simple and seems like effective. Experiments are extensive.

While the regularization term is motivated by analyzing DAN, it would be nice to discuss its application to other domain adaptation/generalization methods. What is even better is to show its effectiveness on another method in the experiments.

---

> ### Author Response · Authors · 2018-11-14
> **Response to Reviewer 1**
>
> Thank you for your positive review.
> We really appreciate the remarks that our “idea is well motivated” and “experiments are extensive and supportive”.
>
> ### Reply to “to discuss its application to other domain adaptation/generalization methods and to show its effectiveness on another method in the experiments” would improve the paper.
>
> Thank you for the suggestion. As you pointed out, the regularization term is motivated by analyzing DAN. So we acknowledge that its application to other domain adaptation/generalization methods is not trivial and discussing it would improve the paper. One possible direction is to modify conditional VAE (Louizos+2015) or CrossGrad (Shankar+2018) to make H(d|h) = H(d|y) holds and to use it in domain generalization. These methods have clear and tractable data generating process but assume the independence of y and d, so we hope we can somehow modify it to H(d|h) = H(d|y) holds. However, we unfortunately might not have time to develop concrete methods and conduct experiments, so we have added the above discussion to the paper.
>
> ### References
>
> Christos Louizos, Kevin Swersky, Yujia Li, Max Welling, and Richard S. Zemel. The variational fair autoencoder. ICLR2016
> Shiv Shankar, Vihari Piratla, Soumen Chakrabarti, Siddhartha Chaudhuri, Preethi Jyothi, and Sunita Sarawagi. Generalizing across domains via cross-gradient training. ICLR2018

---

> > ### Comment · AnonReviewer1 · 2018-11-27
> > **Thanks for the discussions**
> >
> > I wanted to thank the authors for the discussions about VAE and CrossGrad. It would be better to include the same discussions for future work in the manuscript. I have also read the other two reviewers' comments along with the authors' responses. My rating remains positive.

---

### Official Review · AnonReviewer2 · 2018-11-04
**This paper lacks sufficient novelty**

**Rating:** 4
**Confidence:** 5

**Review:**

In this paper, the author(s) propose a method, invariant feature learning under optimal classifier constrains (IFLOC), which maintains accuracy while improving domain-invariance. Here is a list of suggestions that will help the author(s) to improve this paper.
1.The paper explains the necessity and effectiveness of the method from the theoretical and experimental aspects, but the paper does not support the innovation point enough, and the explanation is too simple.
2.In this paper, Figure3-(b) shows that the classification accuracy of IFLOC-abl method decreases a lot when γ is taken to 0. Figure3-(c) shows that the domain invariance of IFLOC-abl method becomes significantly worse when γ is 10. The author(s) should explain the reasons in detail.
3. The lack of analysis on domain-class dependency of each dataset makes the analysis of experimental results weak.

---

> ### Author Response · Authors · 2018-11-14
> **Response to Reviewer 2 [1/2]**
>
> Thank you for your critical feedback.
> We hope to clarify and address your concerns and questions.
> We respond in detail to each comment below.
>
>
> ### Reply to comment 1
>
> Regarding this comment, we would very much appreciate if you could give us more details so that we could exactly address your concerns and improve the paper.
> For example, we would like to know (1) why you think "the paper does not support the innovation point enough" and (2) which parts of "the explanation is too simple" and why simple is a weak point.
>
> Here, we would like to clarify three innovation points of the paper we think.
> (1) We elaborate on the trade-off problem under domain-class dependency, both theoretically (Sec.4.1.1) and experimentally (1st-paragraph of Sec.5.4), for the first time in domain generalization context.
> Here, note that domain generalization is different from domain adaptation in that we cannot obtain input and label data from target domain(s), but has been attracting considerable attention in recent years.
> (2) We propose to maximize domain-invariance within a range that does not interfere with classification accuracy and provide the theoretical analysis which derives the novel approach (i.e., to regularize latent representation so that H(d|y)=H(d|h) holds) to address the aforementioned problem in Sec.4.1.2.
> (3) We confirm the efficacy of the proposed approch with the novel algorighm (IFLOC) in Sec.5.4.
>
>
> Moreover, we have added the below contents to the updated paper in order to clarify the inovation points.
>
> 1. To make it clear that domain generalization differs from domain adaptation, we have added the following sentence:
> ```
> (Sec.2, para.2)  Domain generalization has been attracting considerable attention in recent years (Blanchard et al. (2011); Muandet et al. (2013); Shankar et al. (2018)). Note that it is different from domain adaptation in that we cannot obtain input and label data from target domain(s).
> ```
>
> 2. To make it clear that domain-class dependency (p(y|d) \neq p(y)) is a novel and important problem setting in domain generalization, we compare it with conditional probability shift (p(y|x) and p(x) change across domains) and showed that domain-class dependency is a root cause of the trade-off problem:
> ```
> (Sec.1, para.3) Domain-class dependency might be similar to the situation where p(y|x) and p(x) change across domains due to the causal structure y → x (Zhang et al. (2013); Gong et al. (2016) in domain adaptation and Li et al. (2018c) in domain generalization), which we call conditional probability shift. However, the shift does not cause the trade-off as long as y and d are independent (Figure 1-left), so it is necessary to focus on the relationship between y and d.
> (Sec.2, para.5) There are several kinds of distributional shifts other than domain-class dependency, such as conditional probability shift. Although the distinction between that shift and domain-class dependency is important, it has been received less attention. For example, Li et al. (2018c) claimed that conditional probability shift might harm the performance of domain-invariance-based methods, but our analysis in Sec.4.1.1 suggests that the root cause of the performance degradation is not it but domain-class dependency.
> ```

---

> > ### Author Response · Authors · 2018-11-14
> > **Response to Reviewer 2 [2/2]**
> >
> >
> > ### Reply to comment 2.
> >
> > #### About Figure3-(b):
> > We acknowledge that we lack the explanation on it, so we investigated and considered the reasons as below.
> >
> > (1) Firstly, we do not intend to claim that "IFLOC and IFLOC-Abl always outperform DAN when γ becomes strong", but we just observed that IFLOC and IFLOC-Abl tends to outperform DAN when γ = 10, i.e., the regularizer becomes strong, in Figure 2-(a, b) and 3-(a, b).
> > We provided one possible explanation of this observation that "the regularizers of IFLOC and IFLOC-Abl is KLD and thus bounded by 0, in contrast to that of DAN that can increase to infinity and destabilize the traininig".
> > However, since the regularization terms of IFLOC-Abl and DAN are intended to achive the same equilibrium and they do not have clear superiority or inferiority in theory, we do not intend to exclude the possibility that DAN outperforms IFLOC-Abl on a certain hyperparameter (and of course to show the superiority of IFLOC-Abl over DAN is not the main purpose of the paper).
> >
> > (2) Secondly, we acknowledge that "the classification accuracy of IFLOC-abl method decreases a lot when γ is taken from 0.1 to 1".
> > However, the range of the accuracy reduction is roughly the same with those in <DAN, γ changes from 1 to 10, Figure 3-(a)> and (2) <DAN, γ changes from 1 to 10, Figure 3-(a)>.
> > This suggests that the range of accuracy reduction you concern is not so exceptional for the two domain-invariance-based methods in the WISDM experiment.
> >
> >
> > Based on the above discussions, we modified the paper to clarity that "IFLOC and IFLOC-Abl tends to but not always outperform DAN when γ becomes strong" as follows:
> > ```
> > The training of IFLOC tends to be more stable than that of DAN when the regularizer becomes strong. Figures 2-(a,b) and 3-(a,b) show that IFLOC and IFLOC-Abl could achieve higher Y-Acc than DAN when γ = 1 or 10, i.e., the regularization is strong, except for IFLOC-Abl with γ=1 in Figure 3-(b).
> > This tendency might be because the regularizer of IFLOC is KLD and thus bounded by 0, in contrast to that of DAN that can increase to infinity and destabilize the traininig.
> > ```
> >
> >
> > #### About Figure3-(c):
> > We agree that this should be explained, and we already explained it to some extent as follows:
> > ```
> > (Sec. 5.4, 2nd para.) That high D-Acc might be because the validation accuracy achieved the highest value before the domain-invariance matured. (Recall that the more the representation becomes invariant, the lower the accurary becomes under the trade-off).
> > ```
> > So we would appreciate if you could specify what kind of analysis would improve the paper more.
> >
> >
> > ### Reply to comment 3.
> >
> > We agree that “analysis on domain-class dependency of each dataset” is important, and some analysis is already done in the paper.
> > So we would very appreciate if you could specify what kind of analysis would improve the paper more.
> > Below are the analyses we already did in the paper.
> >
> > 1. BMNISTR is the synthetic dataset which we created by modifying MNISTR. As noted in Sec.5.1, BMNISTRs have several types of domain-class dependency, and the reason for the each type is noted in Sec.5.1. Also, the performances of IFLOC on each dependency type are discussued in the 1st-paragraph in Sec.5.4.
> >
> > 2. PACS has the dependency probably because samples in some <domain, class> pairs are difficult to obtain, as noted in Sec.5.1. The reasons why some <domain, class> pairs become difficult to obtain is discussed in Sec.1 (data characteristics and data-collection errors).
> >
> > 3. WISDM has the dependency due to the reason discussed in Sec.1.
> >
> >
> > Also, we have added the below discussion regarding the datasets.
> >
> > 1. We have added the table of the concrete sample sizes for each <domain, class> pair in PACS and WISDM dataset to Appendix.
> >
> > 2. The reason why WISDM has the dependency was already noted in Sec.1, but we have added it to Sec.5.1 again.
> >
> > 3. We have provided the concrete example of domain-class dependency in PACS as follows:
> > ```
> > (Sec.5.1, para.2) For example, p(y = person|d = Phot) is much higher than p(y = person|d = Sketch), which indicates that photos of person are easier to obtain than those of animals, but sketches of persons are more difficult to obtain than those of animals in the wild.
> > ```
> >
> > 4. We have cited Zhang+2013 to support the fact that that domain-class dependency often happens in real-world dataset as follows:
> > ```
> > (Sec.1, para.4) Unfortunately, domain-class dependency is common in real-world datasets as shown in Zhang et al. (2013).
> > ```

---

### Meta-Review · Area_Chair1 · 2018-12-14
**Potentially interesting new problem setup, but lacking sufficient evidence to showcase problem relevance**

**Confidence:** 4
**Recommendation:** Reject

**Metareview:**

This paper proposes a new solution to the problem of domain generalization where the label distribution may differ across domains. The authors argue that prior work which ignores this observation suffers from an accuracy-vs-invariance trade-off while their work does not.

The main contribution of the work is to 1) consider the case of different label distributions across domains and 2) to propose a regularizer extension to Xie 2017 to handle this.

There was disagreement between the reviewers on whether or not this contribution is significant enough to warrant publication. Two reviewers expressed concern of whether 1) naturally occurring data sources suffer substantially from this label distribution mismatch and 2) whether label distribution mismatch in practice results in significant performance loss for existing domain generalization techniques. Based on the experiments and discussions available now the answer to the above two points remains unclear. These key questions should be clarified and further justified before publication.